

# Fennec dust forecast intercomparison over the Sahara in June 2011

J.-P. Chaboureau[1], C. Flamant[2], T. Dauhut[1], C. Kocha[2,a], J.-P. Lafore[3],
C. Lavaysse[2,b], F. Marnas[2,c], M. Mokhtari[3,d], J. Pelon[2], I. Reinares Martínez[1],
K. Schepanski[4,e], and P. Tulet[5]

[1]Laboratoire d'Aérologie, Université de Toulouse, CNRS, UPS, Toulouse, France
[2]Sorbonne Universités, UPMC Université Paris 06, CNRS and UVSQ, UMR 8190 LATMOS,
Paris, France
[3]CNRM, Météo-France-CNRS, Toulouse, France
[4]School of Earth and Environment, University of Leeds, Leeds, LS2 9JT, United Kingdom
[5]LACy, Université de la Réunion, Météo-France, UMR8105, CNRS, Saint-Denis de La Réunion,
France
[a]now at: MeteoConsult, Paris, France
[b]now at: European Commission, Joint Research Centre, 21027 Ispra Varese, Italy
[c]now at: Capgemini Technology Services, Toulouse, France
[d]now at: Office National de la Météorologie, Algiers, Algeria
[e]now at: Leibniz Institute for Tropospheric Research, Permoserstr. 15, 04318 Leipzig, Germany

*Correspondence to:* J.-P. Chaboureau (jean-pierre.chaboureau@aero.obs-mip.fr)

**Abstract.** In the framework of the Fennec international programme, a field campaign was conducted
in June 2011 over the western Sahara. It led to the first observational data set ever obtained that
documents the dynamics, thermodynamics and composition of the Saharan atmospheric boundary
layer (SABL) under the influence of the heat low. In support to the aircraft operation, four dust

forecasts were run daily at low and high resolutions with convection-parameterizing and convection-
permitting models, respectively. The unique airborne and ground-based data sets allowed the first
ever intercomparison of dust forecasts over the western Sahara. At monthly scale, large Aerosol
Optical Depths (AODs) were forecast over the Sahara, a feature observed by some satellite retrievals
but mislocated by others over the Sahel. The AOD intensity was correctly predicted by the high-

resolution models while being underestimated by the low-resolution models. This was partly because
of the generation of strong near-surface wind associated with thunderstorm-related density currents
that could only be reproduced by models representing convection explicitly. Such models yield to
emissions mainly in the afternoon that dominate the total emission over the western fringes of the
Adrar des Iforas and Aïr Mountains in the high-resolution forecasts. Over the western Sahara, where

the harmattan contributes up to 80 % of dust emission, all the models were successful in forecasting
the deep well-mixed SABL. Some of them, however, missed the large near-surface dust extinction
generated by density currents and low-level winds. This feature, observed repeatedly by the airborne
lidar, was partly forecast by one high-resolution model only.





## 1 Introduction

The Sahara is the world largest desert. In summertime, a large thermal low develops in the lower troposphere over the western Sahara in response to the radiative warming of the surface (Lavaysse et al., 2009). This Saharan heat low plays a pivotal role in the atmospheric regional circulation (e.g. Peyrillé and Lafore, 2007; Chauvin et al., 2010; Martin and Thorncroft, 2014; Lavaysse et al., 2015). The Sahara is also known as the largest source of dust, making dust a key element of the Saharan climate (Knippertz and Todd, 2012). In a recent study, Lavaysse et al. (2011) noticed that the increase in Aerosol Optical Depth (AOD) often coincided with the installation of the heat low over the western Sahara. Based on experiments on sensitivity to dust load, they demonstrated the increase in the thickness of the heat low with dust. In addition to this direct radiative impact, they found an indirect effect, the intensification of the circulation in the Sahel, i.e. the monsoon flow and African Easterly Waves (AEWs). Such remote impact of dust on the regional circulation has already been observed (e.g. Tompkins et al., 2005; Chaboureau et al., 2007; Rodwell and Jung, 2008).

The activation of dust is mainly controlled by soil conditions, surface characteristics and surface wind speed. Because of the persistence of surface characteristics (including dry soil conditions) over the Sahara, the main factor that influences the dust activation here is the low-level dynamics. In the recent years, several mechanisms have been identified, among which are some synoptic features like the harmattan wind, Sharav cyclones (Bou Karam et al., 2010), AEWs (Knippertz and Todd, 2012) and the nocturnal monsoon flow (Bou Karam et al., 2008). Some other mechanisms of dust activation are mesoscale processes, in particular the breakdown of the nocturnal low-level jet in the morning (Todd et al., 2008) and the density currents formed by deep convection in the afternoon (Flamant et al., 2007; Knippertz et al., 2007). The recent capability to run high-resolution models over a large domain has led to process-oriented studies on the role of moist convection in mobilizing dust (e.g., Crumeyrolle et al., 2008; Tulet et al., 2010; Kocha et al., 2013). Strong wind gusts at the leading edge of cold pools are known to generate dust. However, their role is a matter of debate because of the difficulties in detecting them by satellite in the presence of large cloud anvils, and of the inability of low-resolution models to generate such convective systems and their associated mesoscale circulations (see the review by Knippertz and Todd, 2012).

An investigation of the characteristics of the dust properties in the Saharan heat low region was one of the motivations of the Fennec programme. Fennec is an international programme aiming at a better understanding of the Saharan climate system (Washington et al., 2012). In June 2011, a field campaign was organized to measure key properties (i.e. the dynamics, thermodynamics and composition) of the Saharan atmosphere. Two aircraft were operated over northern Mauritania and Mali (Ryder et al., 2015) and ground-based observations were made over Zouerate, Mauritania (Todd et al., 2013) and Bordj Mokhtar, Algeria (Allen et al., 2013; Marsham et al., 2013). A descrip-



tion of the airborne observations together with an overview of the main Fennec results obtained so
far is given by Ryder et al. (2015).

An aspect of the Fennec programme not described so far is the numerical effort to forecast dusty
conditions for the guidance of aircraft. During the 2011 field campaign, four sets of dust forecasts
were specifically designed for Fennec. They were produced every day using three different models.

Two dust forecasts were made with models running with grid spacing of 24 and 20 km. With such a
grid spacing, convection is a subgrid-scale process for which the grid-scale effects are expected to be
represented by a convection parameterization. The two other dust forecasts were made with models
running with grid spacing of 5 km. In these forecasts, no parameterization was used for deep convec-
tion, which was permitted to be explicitly represented. This allowed these high-resolution models

to forecast the density currents associated with thunderstorms while the low-resolution models were
more apt to miss these features crucial for dust emission.

The paper presents an intercomparison of the dust forecasts made for the 2011 Fennec field cam-
paign. The objectives of this intercomparison were to look for any potential systematic error in the
forecasts, to identify the wind regimes leading to dust emission, to estimate their relative contribu-

tion to the total dust emission over the western Sahara and to evaluate the ability of the models to
reproduce the key processes for mobilization and transport. The dust evaluation was made using two
key variables: the AOD retrieved from satellite and ground-based observations and the dust extinc-
tion coefficient derived from airbone lidar measurements. The structure of the Saharan atmospheric
boundary layer (SABL) was also assessed using dropsondes launched over the western Sahara, i.e.

in northern Mauritania and northern Mali.

The paper is organized as follows. Section 2 describes the models and the observations. Section
3 provides an overview of the model performance. Section 4 presents a discussion on the dominant
wind regimes leading to dust emission. Section 5 gives an assessment of the model skill over the
western Sahara where the heat low was installed in the second half of June. Section 6 concludes the

paper.

## 2   Models and observations

### 2.1   Models

The four sets of dust forecasts compared here were performed with three limited-area models, AL-
ADIN (Aire Limitée Adaptation Dynamique INitialisation, Bubnová et al. (1995)), AROME (Ap-

plications of Research to Operations at MEsoscale, Seity et al. (2011)) and Meso-NH (Lafore et al.,
1998). ALADIN and AROME are two spectral models used operationally for weather predic-
tion by Météo-France and other national weather services. They share the same semi implicit,
semi Lagrangian advection scheme as the European Centre for Medium-Range Weather Forecasts
(ECMWF) Integrated Forecast System (IFS) and ARPEGE (Action de Recherche Petite Echelle



Grande Echelle, Courtier et al. (1994)). Meso-NH is a research, grid-point model using a fourth-order centred advection scheme for the momentum components and the piecewise parabolic method advection scheme for other variables.

ALADIN is a hydrostatic model while AROME and Meso-NH are non-hydrostatic. All these three models use the same parameterizations for surface processes and radiation, that is, the Interactions

between Soil, Biosphere and Atmosphere (ISBA) surface scheme (Noilhan and Planton, 1989), the Rapid Radiative Transfer Model (RRTM) parameterization (Mlawer et al., 1997) for longwave and the two-stream formulation originally employed by Fouquart and Bonnel (1986) for shortwave. In addition, AROME and Meso-NH share their physical parameterizations for microphysics, turbulence and shallow convection. More details on ALADIN, AROME and Meso-NH can be found in

Mokhtari et al. (2012), Kocha et al. (2012) and Chaboureau et al. (2011), respectively.

The limits of the model domains are shown in Fig. 1. The grid spacing was 24 km for ALADIN and 5 km for AROME. For these two models, initial and boundary conditions were taken from operational large-scale ARPEGE forecasts at 18:00 UTC. ALADIN and AROME were integrated forward for 72 and 48 h, respectively. For Meso-NH, two grid spacings were employed, 20 and

5 km leading to two sets of forecasts, named MNH20 and MNH05 respectively hereafter. These two grid configurations covered the same domain as shown in Fig. 1. Meso-NH was initialized by the ECMWF analysis at 00:00 UTC and run for 24 h using the ECMWF forecasts for the lateral boundary conditions. All the forecasts started from aerosol-free conditions. Then, dust prognostic variables at the end of a given 24 h forecast are were passed on as initial conditions at the start of

the next 24 h forecast. Model outputs were saved every 3 h, some diagnostics being missed for a few models. The comparison examines the first day of forecasts, i.e. between lead times of 9 and 30 h for ALADIN and AROME and 3 and 24 h for Meso-NH. Because of the 5 km grid spacing, explicit deep convection was permitted in AROME and MNH05 and no parameterization was employed for subgrid deep convection.

The three models share the same dust prognostic scheme described in Grini et al. (2006). Dust fluxes are calculated from wind friction velocities using the Dust Entrainment and Deposition (DEAD) model (Zender et al., 2003). The physical basis of the model is taken from Marticorena and Bergametti (1995) in which dust fluxes are calculated as a function of saltation and sandblasting processes. The horizontal saltation flux is first calculated based on the wind fric-

tion speed. Then, the dust vertical flux equals the total horizontal saltation mass flux weighted by the sandblasting efficiency. The dust emissions are forced directly by the surface flux parameters of the ISBA surface scheme, and then distributed into the atmosphere. In this parameterization, the three lognormal modes are generated and transported by the lognormal aerosol scheme of the OR-ganic and Inorganic Log-normal Aerosols Model (ORILAM, Tulet et al., 2005). The initial dust size

distribution contains three modes with median diameters of 0.078, 0.64 and 5.0 μm and standard deviations of 1.7, 1.6 and 1.5, respectively as defined by Crumeyrolle et al. (2011). Dust loss occurs



through sedimentation and rainout in convective clouds. Radiative properties of dust are calculated within ORILAM, which is coupled on-line to the radiation scheme.

The three models used different versions of DEAD. ALADIN used the version improved by Mokhtari et al. (2012) to better take the size distribution of erodible material into account. In this version, the surface soil size distribution depends on the soil texture following Marticorena and Bergametti (1995) and the sandblasting efficiency is parameterized following Shao et al. (1996). AROME and Meso-NH used the original version of DEAD. The surface soil size distribution was assumed to be uniform without any restriction on the number of particles available for saltation. The sandblasting efficiency was calculated following Marticorena and Bergametti (1995) in which the sandblasting efficiency depends on the clay fraction available in the soil (up to a limit of 20 %). AROME and Meso-NH also differed by a mass tuning factor used to estimate the horizontal saltation flux. This factor depends on the dynamics and grid spacing of the meteorological model. Therefore, it needs to be tuned. Its value was $1.6 \times 10^{-3} \, \mathrm{cm}^{-1}$ in MNH20, $1.0 \times 10^{-3} \mathrm{cm}^{-1}$ in MNH05 and $15 \times 10^{-3} \mathrm{cm}^{-1}$ in AROME.

## 2.2 Observations

The forecast intercomparison took advantage of the unique observation data set obtained from the SAFIRE (Service des Avions Français Instrumentés pour la Recherche en Environnement) Falcon 20. The aircraft was equipped with the LEANDRE Nouvelle Génération (LNG) backscatter lidar (Bruneau et al., 2015). The profiles of atmospheric extinction coefficient at 532 nm were retrieved using a standard lidar inversion method that employs a backscatter-to-extinction ratio of $0.0205 \, \mathrm{sr}^{-1}$ (see Schepanski et al. (2013) for a more detailed description of the inversion method). The retrievals had an estimated uncertainty of 15 %, a resolution of 2 km in the horizontal and 15 m in the vertical. The Falcon 20 was also equipped with a dropsonde capability. It was based at Fuerteventura (Canary Islands, Spain) from 1–23 June 2011. The aircraft flew over the western African coast during the maritime phase (2 to 12 June, Todd et al. (2013)) and over the Sahara during the heat low phase (13 to 30 June). When the aircraft was cruising, the flight altitude was maintained constant at approximately 11 km above mean sea level. The reader is referred to Ryder et al. (2015) for a comprehensive description of the instrumentation on board the Falcon 20 as well as the complete list of the flights.

The forecast intercomparison also took advantage of the Aerosol Robotic Network (AERONET) stations located in the Sahara (Zouerate, Bordj Mokhtar and Tamanrasset, Fig. 1). It is worth noting that the AERONET stations at Zouerate and Bordj Mokhtar were installed specifically for the 2011 Fennec field campaign. We used AOD at 500 nm and the 440–870 nm Ångstrom exponent from the three stations. For the purpose of comparison with three-hourly forecast outputs, the AERONET products were averaged within a time-window of 3 h. The Ångstrom exponent characterizes the AOD dependence on wavelength and provides information on the aerosol type and size. The purely dust





cases correspond to the lowest values of the Ångstrom exponent. In general, dust is characterized by low values of Ångstrom exponent, less than 0.4.

To obtain an assessment of the models performance at the regional scale, we used satellite-based AOD retrievals from the Moderate Resolution Imaging Spectroradiometer (MODIS) and the Multi-angle Imaging SpectroRadiometer (MISR). The MODIS AOD at 550 nm was obtained from the Deep Blue collection 5.1 over bright surfaces such as deserts (Hsu et al., 2006). Comparisons of AOD between the Deep Blue algorithm and AERONET sun photometers showed a general agreement within 20–30% for AOD (Hsu et al., 2006). We used the MODIS instrument on board Aqua, which crosses the equator at around 13:30 local time. (Aerosol Deep Blue products were not available for Terra after December 2007 because of unavailability of the required polarization corrections to the radiance data.) The MISR AOD used here is the monthly AOD average (level 3 product) taken in the green band (558 nm). MISR is aboard Terra, which crosses the equator on the descending node at about 10:30 local time. A global comparison of MISR and AERONET AODs showed that 63% of the MISR AODs fell within 0.05 or 20 % of AERONET AODs (Kahn et al., 2005). Products were on a 1° latitude-longitude grid for MODIS and 0.5° latitude-longitude grid for MISR. The global AOD coverage with MISR was achieved in 9 days due to its narrow swath of 360 km while MODIS with a 2400 km wide swath achieved a near-global AOD coverage on a daily basis. As a result, the mean number of retrievals for June 2011 was 28 for MODIS and 5 for MISR.

## 3 Assessment of AOD

### 3.1 Overall evaluation against satellite observations

We first provide a comparison of monthly mean forecasted AODs at 500 nm for June 2011 against equivalent quantities derived from satellite observations (Fig. 2). The forecasted AODs were averaged at 12:00 UTC, the closest output time to the observations. Note that the dust load follows a diurnal cycle over West Africa (Chaboureau et al., 2007). The single daytime observation from either the Aqua or the Terra satellite led to an under-sampling of the dust cycle. This under-sampling, combined with the effect of the cloud masking, strongly affected the reliability of the AOD. For MODIS retrievals, Kocha et al. (2013) estimated an AOD underestimation of 0.28 over the convective regions and an AOD overestimation of 0.1 over morning source areas.

MODIS showed an AOD maximum of 1 in the Bodélé depression over northern Chad, known as one of the most intense sources of dust in the world. This area of large AOD extends westward over the Erg of Bilma (northeastern Niger). Large AOD values were also found by MISR in approximately the same locations, but with different magnitudes (e.g., 1 for MISR against 0.6 for MODIS over the Erg of Bilma, 18° N/14° E). From the southern flanks of Adrar des Iforas to the Atlantic coast, the two retrievals differed strongly in the meridional location and magnitude of the AOD maximum. It was located over the Sahel for MODIS (values around 0.7 between 14° and 16° N) and



over the Sahara for MISR (over 1, between 18° and 20° N). Another area of disagreement was the northwestern edge of the Hoggar Mountains and the Tademaït plateau (28° N/2°E), where MODIS AOD retrievals were larger than 0.5 while MISR showed values less than 0.4. However, MODIS
and MISR agreed on the AOD range (between 0.3 and 0.5) over the Grand Erg Occidental between the Hoggar and Atlas Mountains. In a comparison of satellite observation of Saharan dust source areas, Schepanski et al. (2012) found large discrepancies between MODIS and the Spinning Enhanced Visible and Infra-Red Imager (SEVIRI). As in Fig. 2a, Schepanski et al. (2012) found the large frequency of MODIS AOD larger than 0.5 in the Sahel region. Their result contrasted with the
dust source activation derived from SEVIRI for which a large strip of activation was found along the southern Algerian border with Mali and Mauritania. By contrast, MISR shows higher AODs further north, particularly over the central Sahara, in better agreement with the SEVIRI-derived source regions. This is likely related to the overpass time of MISR which is closest to the time of source activation as the result of the breakdown of the nocturnal low-level jet in the morning.

All the forecasts agreed with each other in showing a strip of large AOD around 18° N, consistently with MISR, but in disagreement with MODIS. The forecasts differed significantly, however, in the range of AOD. ALADIN showed the lowest value around 0.4, a little lower than MNH20 with values around 0.6. The two convection-permitting forecasts exhibited much higher values, up to 1, which matched the largest MISR values. The difference in MNH20 and MNH05 suggests that these
high AOD values were caused by processes better represented at high resolution. This concerned wind acceleration by both topographical channeling and dust uplift at the leading edge of density currents related to thunderstorms. The convection-permitting models also delivered the best forecasts by capturing the meridional gradient of AOD observed by MISR over the western Sahara (the rectangle in Fig. 2 delimits the area within which most of the aircraft operation took place in June
2011, see Ryder et al. (2015)). There, the average AOD was the lowest for ALADIN (0.3) and the highest for MNH05 (0.7).

    Over the Grand Erg Occidental, MNH20 and MNH05 showed much larger AODs than ALADIN and AROME, with AOD values of 0.7 for Meso-NH against 0.4 for the other two models. The contrast between these two sets of forecasts has to be attributed to wind speed (as discussed in the
next section), in relationship with the difference in the dynamical core and/or the initial conditions provided by different global models. Note that the models with the smallest domain (AROME and Meso-NH) show a lack of AOD at the eastern boundary. The reduced coverage by these models can be a strong limitation as dust can travel from Sudan to West Africa (Flamant et al., 2009b). Lastly, the maximum over the Bodélé depression appears to be missed by all the models. This is also
true for ALADIN despite the corrections implemented in the revised DEAD version. This failure could be attributed to an underestimation of near-surface wind speed as previously noted in a model intercomparison study dedicated to Bodélé (Todd et al., 2008).




### 3.2 Comparison of AOD at AERONET stations

We now consider comparisons of forecasted AOD against AERONET AOD at 500 nm using time
series from the three stations located in the Sahara (Fig. 3). The corresponding MODIS values are
also shown, together with the 440–870 nm Ångstrom exponent from AERONET. As described by
Todd et al. (2013), June 2011 can be separated in two distinct periods with different meteorological
conditions. In the first period, the so-called maritime phase from 2 to 12 June, the western Sahara
was under the influence of synoptic disturbances originating from the Atlantic while the heat low
was located around 15° E. During the second period, the so-called heat low phase, from 13 to 30
June, the heat low migrated in a westward location at 5°–10° W and a series of AEWs propagated
across the western Sahara.

At Zouerate (Fig. 3a), the change in AOD was particularly representative of the westward migra-
tion of the heat low in the western Sahara. During the maritime phase (2–12 June), AOD values were
lower than 0.2 while the Ångstrom exponent shows values larger than 0.4. Because of the prevalence
of northwesterlies, these low values of AOD were probably due to sea salts coming from the Atlantic.
Near-zero values of AOD were correctly forecast by all the models while the MODIS retrieval over-
estimated AOD strongly with values around 0.4. From 13 June onwards, Zouerate was primarily
affected by dust advecting with the northeasterlies (Todd et al., 2013). AERONET AOD increased
in the range between 0.6 and 1.2 while the Ångstrom exponent was below 0.4. This AOD increase
was retrieved from MODIS and forecast rather well by the models. AERONET AOD peaked at 1.2
on 21 and 23 June. AROME, MNH20 and MNH05 regularly forecast peaks of AOD, but for different
days and with larger magnitude than observed with AERONET (up to 1.9, 1.9 and 1.3, respectively).
MODIS and ALADIN presented much less temporal variation, AOD being limited to a maximum of
1.0 and 0.8, respectively.

At Bordj Mokhtar (Fig. 3b), the AOD was characterized by an enhanced variability of AOD with
respect to Zouerate. The Ångstrom exponent remained below 0.4, indicating that dust contributed
the most to the AOD. Before 8 June, the centre of the heat low was close to the station, to its west.
Consequently, the station was affected by the moist south-westerly monsoon flow and the AOD
was observed to exceed twice the value of 1 (Marsham et al., 2013). The high-resolution forecasts
presented a similar change in AOD, but not at the right time. Between 8 and 12 June, the heat low
moved to its easternmost position and did not significantly influence the AOD observed in Bordj
Mokhtar. In the absence of any significant meteorological disturbance affecting the station, the AOD
decreased below 0.4. The low value of AOD was well forecasted by all the models. From 13 June
onward, the station was inside the heat low as it settled in its Saharan location for the year 2011. Two
dust emission processes were predominant: cold pools associated with deep convection initiating in
the southerly monsoon flow and the northeasterly harmattan on the western flank of the heat low
(Marsham et al., 2013). The AOD was frequently observed to exceed the background value of 0.4.
It sometimes reached values as large as 2. Five episodes of AOD larger than 1 and lasting for about





three days were distinguished on 13, 17, 21, 25 and 29 June associated with propagating AEWs. AEWs favour dust emission by increasing the low level flows and conditions creating conditions where deep convection tends to occur. The occurrence of these high AODs was better simulated in the high-resolution forecasts. This shows the predominant role of deep convection at the origin of these AOD peaks.

At Tamanrasset (Fig. 3c), the AOD derived from the AERONET station was always above 0.4 and the Ångstrom exponent less below 0.4 until 20 June. There were four episodes with AODs above 1 associated with near-zero values of Ångstrom exponent. These days corresponded to episodes of long-range transport of dust from eastern sources, some of which were associated with the same AEWs than those travelling across Bordj Mokhtar, rather than local emissions. It is worth noting that

most of the Hoggar Mountains is covered with silt loam in which the low percentage of sand does not favour saltation and hence dust emission. The increase in Ångstrom exponent seen in late June suggests that dust was mixed with other aerosol species. All the forecasts gave an AOD temporal variability that was lower than observed. However, two dust episodes (on 15 and 19 June) out of four were forecast correctly in time, but not in magnitude, the forecasted AODs being biased towards

low values. The low bias AOD was largely explained by an underestimation of the transport from eastern sources as well as the lack of remobilization of dust from the eastern sources deposited on the surface, which is not represented by the DEAD scheme, but is frequently observed in this region.

   The comparison of AOD time series is summarized in a quantitative way by means of a Taylor diagram in Fig. 4, where the AERONET AOD values at the three stations are taken as references.

Overall, the forecasts show rather similar skills for all AERONET stations. In most of the cases, they underestimate AODs with a bias between $-40$ and $-60\,\%$. A notable exception is the bias at Zouerate, which is smaller than $-20\,\%$ for ALADIN and AROME and $+8$ and $+29\,\%$ for MNH05 and MNH20, respectively. Relatively good scores in terms of correlation were achieved at Zouerate and Tamanrasset, the stations that bordered the heat low in the second half of June 2011. The lowest

correlation coefficients (between 0.21 and 0.41) were obtained in Bordj Mokhtar, where the AOD variation found was the largest, thus the most difficult to forecast, especially regarding deep convection and related dust uplifts at the leading edge of the cold pools. There was no a strong contrast in scores between forecasts initialized with ARPEGE and ECMWF nor in forecasts at low and high resolutions.

It is worth noting that using MODIS AODs instead of forecasts did not improve the scores. At Zouerate, MODIS was high-biased against AERONET ($+23\,\%$) and showed the smallest correlation coefficient (0.65). This was due to the overestimated AOD during the maritime phase. At Tamanrasset and Bordj Mokhtar, MODIS was biased particularly low against AERONET, with biases as low as $-72$ and $-67\,\%$, respectively. MODIS presented the highest value of the correlation coefficient at

Tamanrasset (0.69) but the lowest at Bordj Mokhtar (0.21). This makes the use of MODIS retrieval



as a reference for model validation questionable. It also points out the great advantage of installing two AERONET stations specifically at Zouerate and Bordj Mokhtar during the Fennec campaign.

## 4 Wind regimes controlling dust emission

Dust emission probably explains most of the differences in AOD among the models. In addition to the changes made in the revised version of DEAD, the difference in emission was due to differences in initial conditions (from either ARPEGE or ECMWF) and model characteristics (advection scheme and grid mesh). Here, dust emission is examined first by looking at the source areas and the wind regimes they experienced and second by focusing on the western Sahara, the area where the Fennec aircraft operation took place. Because of operational constraints, dust emission fields calculated from AROME were not saved and therefore are not included in this analysis.

### 4.1 Attribution of dust emission to wind regimes

Dust emission occurs when wind friction speed exceeds a threshold that depends on the surface roughness and soil moisture. High winds are associated with different wind regimes at synoptic, regional and local scales. Atlantic inflow associated with synoptic systems affects the western coast of northern Africa (Grams et al., 2010). Over the continent, the wind regime in June depends on the location of the intertropical discontinuity (ITD). The ITD is the near surface convergence zone between the moist, cold monsoon flow and the dry, warm harmattan. To the north of the ITD, the harmattan wind blows from the northeast or east in the western Sahara. To the south of the ITD, the monsoon flow is a southwesterly wind. These low-level winds can strengthen during the night when the decoupling between surface and atmosphere occurs. Last but not least, density currents associated with thunderstorms lead to dust emission (see the review by Knippertz and Todd, 2012).

The attribution of wind regimes to dust emission is a two-step method. In order to be easily applicable to models with different horizontal and vertical resolutions, it is based on wind at $10\,\mathrm{m}$ only. In the first step, we define dust emissions as surface objects. This allows us to describe each individual dust emission cluster with the average and standard deviation of their wind speed and direction. A dust emission grid point is part of the same cluster as its neighbours if they share a common face in the horizontal cardinal direction. In other words, diagonal connections are ruled out. In the second step, we attribute the dust emission of a cluster to a wind regime depending on its mean wind direction and its wind speed standard deviation. In case of a standard deviation of wind speed larger than $3\,\mathrm{m\,s^{-1}}$, the emission is attributed to a cold pool. Cold pools associated with thunderstorms emission are local processes that show a much larger spatial variability within a dust emission cluster than large-scale winds. Otherwise, emission was attributed to the monsoon flow or the Atlantic flow or the harmattan. To attribute the emission to one of these three wind regimes, we divided the domain of analysis with respect to the position of the ITD. Here, the ITD was



defined as the southern limit where the mixing ratio of water vapour at 2 m equals $10\,\mathrm{g\,kg^{-1}}$. This value corresponds to the $14°\,\mathrm{C}$ dew-point temperature criterion used by Flamant et al. (2009a) and Bou Karam et al. (2009) among many others. To the south of the ITD, the emission was attributed to the monsoon flow. To the north of the ITD, the dust emission associated with northwesterly winds west of $10°\,\mathrm{W}$ was attributed to the Atlantic flow. Otherwise, the emission process was attributed to

the harmattan.

     An example of dust attribution to wind regime is shown for 21:00 UTC on 20 June (Fig. 5), the evening before the morning flight F22 (discussed in the next section). Low values of brightness temperatures at $10.8\,\mu\mathrm{m}$ (blue and white colours) were observed by SEVIRI over the Atlas Mountains, southern Mali, between Mauritania and Mali and at the southern tip of the Hoggar Mountains

(Fig. 5a). These low values corresponded to high clouds, typical of anvils of mesoscale convective systems (MCSs). Similar features were simulated by MNH05 (Fig. 5b; see Chaboureau et al. (2008) for further details of the brightness temperature calculation). The MCSs were forecast at about the same locations as observed, showing the good skill of MNH05. Dust emission was associated with the MCSs over Niger and the Atlas Mountains (Fig. 5c). Because the standard deviation of the 10 m

wind speed exceeded $3\,\mathrm{m\,s^{-1}}$, it was attributed to cold pools (Fig. 5d). In other places, dust emission was attributed to some large-scale wind regimes: to the Atlantic inflow over Mauritania, to the monsoon flow over Senegal and to the harmattan elsewhere.

### 4.2   Dust emission sources and prevailing wind regimes

     The magnitude of the dust emission and the dominant wind regimes leading to emission in June

2011 are shown in Fig. 6. Arrows indicate the amplitude and phase of the diurnal variation of dust emission. For the sake of visibility, ALADIN emission was multiplied by 3.

     Emission due to the harmattan was the main mechanism for the three forecasts. It occurred in specific areas, between the Atlas and Hoggar Mountains in Algeria, over Libya and in the Bodélé depression. Over Libya, in the Bodélé depression and some other places, the maximum in the 3-

hourly accumulated emission occurred at 09:00 UTC. This morning maximum corresponded to the break-down of the nocturnal low-level jet and the downward transfer of momentum leading to high near-surface winds. Over central Algeria and northern Mauritania, the emission was maximum between 15:00 and 18:00 UTC in connection with particularly strong harmattan winds and the surface warming, which is maximum over the central Sahara in that time frame. Emission due to the

Atlantic inflow impacted western Africa. Over the coasts, it was maximum at 18:00 UTC. By construction, emission due to monsoon flow was found over the Sahelian band south of $20°\,\mathrm{N}$. The time of the maximum differed among the models. It occurred at 12:00 UTC for ALADIN and MNH20 and 21:00–24:00 UTC for MNH05. The last mentioned early night-time maximum of emission agreed well with the dust emission observed at the leading edge of the nocturnal monsoon flow

(Bou Karam et al., 2008). Finally, emission with cold pools as the dominant mechanism was found



over the western fringes of the Adrar des Iforas and Aïr Mountains for MNH05 only. This agrees well with the location of deep convective systems close to mountains, a well-known feature of the African monsoon (e.g. Söhne et al., 2008; Schepanski et al., 2012; Kocha et al., 2013).

In the DEAD scheme used for all the forecasts, dust emission depends on the horizontal dust flux, which is a function of several parameters, including the cube of the friction velocity. The latter is derived from the 10 m wind speed. It was therefore expected that the largest AOD and emission seen for the high-resolution model forecasts would be explained by a change in wind speed. Note, however, that the sandblasting efficiency, which gave the dust vertical flux from its horizontal counterpart, varies greatly between the models.

The frequency of the 10 m wind speed is shown in Fig. 7 for the four forecasts (including AROME) at every output time. Whatever the model forecast, the frequency of 10 m wind speed drops rapidly to near $7\,\mathrm{m\,s^{-1}}$ below frequencies of less than 5 %. For all the models, the frequency of strong winds is maximum in the afternoon, around 18:00 UTC, and minimum around 09:00 UTC. With the exception of ALADIN, the slope for large wind speed diminishes with time. While wind speed is limited to values around $15\,\mathrm{m\,s^{-1}}$ in ALADIN, it can reach $20\,\mathrm{m\,s^{-1}}$ in MNH20 and above $25\,\mathrm{m\,s^{-1}}$ in AROME and MNH05. On 1‰ of the domain, the wind speed is over $12\,\mathrm{m\,s^{-1}}$ for the low-resolution models while exceeding $15\,\mathrm{m\,s^{-1}}$ in the afternoon for the two high-resolution models. The larger wind speed produced at high resolution was related to a more detailed representation of orography and physical processes. Because this increase occurred in the afternoon, it was related to strong harmattan winds evidenced in Fig. 6 and discussed previously. In the case of the high-resolution models, a fraction of the high wind speeds at 18:00 UTC was also probably due to density currents produced by thunderstorms described explicitly.

**4.3 Diurnal cycle of dust emission over the western Sahara**

Dust emission was further examined by looking at its diurnal cycle over the western Sahara where the aircraft operation took place (Fig. 8). ALADIN presented the lowest dust emission with a daily average of $0.6\,\mathrm{g\,m^{-2}}$. Emission occurred meanly between 09:00 and 18:00 UTC, peaking at around noon while being very low during the night and early morning. MNH20 showed a much stronger daily average of $4.6\,\mathrm{g\,m^{-2}}$. Dust emission also peaked around noon, but the period of large emission started earlier, i.e. extending from 06:00 to 18:00 UTC. Despite stronger near-surface winds than in MNH20, MNH05 emitted 74 % less dust (an average of $3.4\,\mathrm{g\,m^{-2}}$) mainly because the sandblasting efficiency was reduced by 62 %. Compared to the low-resolution forecasts, the emission peak in MNH05 was delayed in the afternoon, i.e. 15:00 UTC instead of 12:00 UTC for MNH20 and ALADIN. The evening emission was also significantly higher than the early morning one for both MNH05 and MNH20, unlike for ALADIN. In MNH05, the evening dust emissions were larger than in MNH20, due to the contribution of cold pools related to evening thunderstorms.



For all the models, and independently of the time considered, the dominant wind regime for dust emission over the western Sahara was the harmattan. In total, it accounted for 80 % in ALADIN, 78 % in MNH20 and 74 % in MNH05. This regime occurred mainly between 09:00 and 18:00 UTC. The second most important wind regime was the Atlantic flow accounting for 17 % in ALADIN and MNH20 and 15 % in MNH05. The third wind regime was the monsoon flow, for 5 % in MNH20 and MNH05 and 4 % in ALADIN, with the strongest values between 09:00 and 12:00 UTC. Finally, emission due to cold pools was diagnosed between 15:00 and 24:00 UTC for MNH05 only. It accounted for 6 % of the total dust emission. It is worth emphasizing that this value was calculated for the western Sahara (i.e. the domain delimited by the box in Fig. 6). Higher values were obtained over other areas, such as the western fringes of the Adrar des Iforas and Aïr Mountains.

## 5 Assessment in the heat low region

The vertical structure of the atmosphere is now assessed in the heat low region using observations obtained from the Falcon 20. In the following, profiles of temperature, moisture and dust extinction from forecasts are first compared with dropsonde and LNG observations during the so-called heat low phase, when the Saharan heat low was moving towards western Sahara. Three flights representative of the meteorological conditions of the heat low phase over the western Sahara are then presented.

### 5.1 Vertical structure of the Saharan atmosphere

We selected the eight long-range flights operated between 14 and 22 June (flights F16, F17, F19, F21, F22, F23, F24, and F25). The objective of F16, F17 and F19 was to document the boundary layer. F21 was dedicated to the survey of dust associated with the ITD and the heat low. F22 was designed to survey dust associated with a Mediterranean surge and density currents from the Atlas Mountains. The purpose of F24 and F25 was to survey the Saharan heat low. For each flight, we only analysed data from the longest leg because they provided the most comprehensive view of the Saharan atmosphere (Fig. 9). During the heat low phase, the averaged wind at 925 hPa (from MNH20) exhibited an almost closed cyclonic circulation over eastern Mauritania and the MISR AOD was larger than 0.7 over a zonal band around 18°–20° N. For the sake of concision, the profiles of potential temperature, relative humidity and dust extinction coefficient were averaged along the flight tracks (Fig. 10). The forecast outputs closest to the time of the flight tracks were selected.

In the afternoon, most of the observed temperature profiles showed a well-mixed, deep SABL (Fig. 10 top). The potential temperature was quasi-uniform with values of 318–320 K in the first 4.5–5.5 km. F19 differed from the other afternoon flights by showing a 1 km layer that was cooler than aloft. This air was advected from the Atlantic as detailed in the next sub-section. The morning





flight F22 showed a cold-air layer associated with northeasterlies. A more detailed analysis of F22
is given in subsection 5.3.

At the surface, the air was very dry with a relative humidity of 5 % (Fig. 10 middle). The air
became more moist with altitude reaching a maximum between 50 and 70 % at the SABL top. The
SABL top reached an altitude of around 5.5 km on 14 and 15 June, then moved lower to 4 km on
16 June due cool, moist near-surface conditions related to the inland penetration of the Atlantic in-
flow which delayed the SABL development. The SABL depth then rose again to 6 km from 21 June
onwards. These changes in the moisture distribution at the SABL top were caused by the passage
of AEWs. The latter enhance the northward propagation of the monsoon flow in the low-levels, this
moisture then being mixed vertically by the strong updrafts in the SABL. In particular, moisture
events were reported on 13, 17 and 21 June at Bordj Mokhtar (Marsham et al., 2013). This matches
the increase in moisture observed further west on 15, 20 and 21 June (F17, F21, and F23, respec-
tively).

The models represented the lower part of the temperature profile rather well (Fig. 10 top). They
reproduced the profile of 14 June (F16) and the afternoon of 22 June (F25) remarkably well, with
maximum differences of less than 2 K below 4 km (with the exception of ALADIN in the first kilo-
metre on 14 June). The difference in the SABL temperature was the largest on 16 and 20 June (F19
and F21), two flights during which cool near-surface conditions were observed in relationship with
an Atlantic outflow and the monsoon flow. The forecasts were successful in capturing the dry air in
the first kilometres (Fig. 10 middle). An exception was F21, which documented the northern fringes
of the monsoon flow. The forecasts of relative humidity at the SABL top were more erroneous.
Meso-NH followed the observations rather well while ALADIN and AROME were either too dry
(F17, F21, F22 and F23) or too moist (F19).

Dust extinction was observed within the SABL only (Fig. 10 bottom). It decreased with altitude,
the largest amount being found close to the surface. Two major dust events were reported with
extinction larger than $0.5\,km^{-1}$ in the first kilometre (flights F22 and F24). They corresponded to
dust uplifts associated with a density current and a harmattan outbreak, respectively (see subsection
5.3 on F22 below). The mean vertical profiles of dust extinction in the forecasts showed systematic
errors. For most of the flights, dust was correctly forecast below the SABL top. There were, however,
some exceptions (F21 for Meso-NH and F17, F19, and F21 for AROME). Another drawback was
excessively large values of extinction in the upper part of the SABL. This can be seen for F17
for all the forecasts. In the lowest levels, ALADIN and AROME forecast reduced dust extinction
wrongly. The structure was much better forecast by Meso-NH. However, some peaks were strongly
underestimated (e.g., F24) or overestimated (F21 by MNH05). This shows the great difficulty of
forecasting dust events accurately at the mesoscale.




### 5.2 Dust mobilization at the installation of the heat low

A first example of the vertical section of dust extinction is taken from the flight conducted on 16 June 2011 at the time of the installation of the heat low in its Saharan location. Figure 11 shows the longest leg of the flight F19, which took place between 15:18 and 16:00 UTC superimposed on the AOD from MODIS, MISR and the dust forecasts at 15:00 UTC. On the western part of the MISR overpass, the satellite retrievals were in agreement. In particular, they both retrieved with

the Zouerate AERONET rvalue of 0.3. The relatively low AOD values over Western Sahara were associated with the cool northeasterly Atlantic inflow. On the eastern part, MISR presented a strip of AOD over 0.6 that was absent in MODIS. The forecasts all showed this strip of enhanced AOD, in better agreement with MISR than with MODIS. In the morning, the south-westerlies mobilized dust over the Western Sahara while the north-easterlies activated the sources of the Grand Erg Occidental

further north. The AOD strip marked the convergence of these two wind flows at low level. The north-easterlies at 925 hPa were stronger in Meso-NH than in ALADIN and AROME, which explains the difference in the magnitude of the forecasted AOD.

Consistently with MISR, the LNG observation showed a strong gradient of dust extinction in the boundary layer across the track (Fig. 12a). Dust extinction larger than $0.05\,\mathrm{km}^{-1}$ was found in a

layer for which the depth increased from $2\,\mathrm{km}$ over northern Mauritania due to the cooler temperatures) to $6\,\mathrm{km}$ in Central Mauritania. It resulted in AODs varying from 0.3 to 1.1 as derived from LNG (Fig. 12b). This increase was reproduced by all models, exception made of ALADIN which exhibited a nearly constant AOD across the domain as seen in Fig. 11c. AROME and Meso-NH exhibited an AOD maximum at a distance between 300 and 400km while LNG observed a contin-

uously increasing AOD. Over Central Mauritania, dust extinction larger than $0.3\,\mathrm{km}^{-1}$ observed in the first kilometre (at a distance between 400 and $500\,\mathrm{km}$) corresponded to a dust uplift associated to the flow coming the Atlantic and merging with the monsoon flow in the southeastern part of the aircraft leg (Fig. 11f). The increase in extinction with the distance from the northeastern position of the leg was forecast by all the models (Fig. 12c-f). All models show the intrusion of colder air in

the lower 2-km associated with the cool Atlantic inflow. However, they all tend to forecast too much dust in the residual layer between 2 and 6-km altitude, over the Atlantic inflow. Nevertheless, the MNH05 forecast evidenced the slanted dust filaments as in the LNG observations (Fig. 12f) while AROME forecast an almost dust-free air in the whole atmospheric column (Fig. 12d), thereby also mimicking the observations. Over Central Mauritania (within distances between 400 and $500\,\mathrm{km}$),

Meso-NH was the only model able to forecast the dust uplift associated with the strong southwesterlies. AROME lacked any significant extinction at the first levels as already seen in the vertical profile for the F19 flight (Fig. 10 bottom).





### 5.3 Dust mobilization due to a Mediterranean surge and overnight density current

A large part of dust emission observed in the second half of June 2011 in the area of interest was due to strong harmattan wind (see Fig. 8). An example is taken here from the leg of flight F22 between 07:58 and 08:50 UTC on 21 June. This case is particularly interesting as additional emission of dust was due to an overnight density current originating from deep convection over the Moroccan Atlas Mountains (Todd et al., 2013).

Along the F22 flight track, the MODIS AOD showed values greater than 0.8 (Fig. 13a). A maximum of AOD larger than 1.2 was even reached in northern Mauritania. These values matched those obtained from LNG rather well (Fig. 14b). Over the Western Sahara, MODIS and MISR AODs differed significantly (less than 0.4 and in excess of 0.8, respectively, Fig. 13a,b). The 2 h delay between the satellite overpasses can hardly explain this discrepancy.

The AOD forecasted by ALADIN at 09:00 UTC was the most different from the two satellite observations (Fig. 13c). It varied between 0.2 and 0.4 over most of the area. In the southwestern corner, AODs larger than 0.6 were forecast to be associated with southerly winds. This area of large AOD expanded further northeast in AROME, which forecast south-westerlies (Fig. 13d). The difference in wind direction might explain the difference in AOD. In Meso-NH, the southwestern area with large AOD spread considerably northward (Fig. 13e,f). It even reached Zouerate where the AERONET station recorded AOD values of 1 in the morning. It also spread into the western Sahara providing the AOD forecasts closest to MISR observations. Along the F22 track and to its northeast, AROME and Meso-NH showed large AOD values that were lacking in ALADIN. The MODIS maximum was well forecast by Meso-NH, but missed by AROME and ALADIN. The MODIS secondary maximum at 6° W/22° N was forecast by AROME, but was missed (or delayed) by Meso-NH and ALADIN.

When the vertical section of dust extinction was examined (Fig. 14), the LNG observation revealed a near-surface dust aerosol layer with extinction larger than $0.4 \, \text{km}^{-1}$ in the first kilometre (Fig. 14a). This layer resulted from a southward Mediterranean surge enriched with an overnight current from thunderstorms triggered over the Atlas Mountains (Todd et al., 2013). ALADIN forecast an increase in dust near the surface, but with too low a magnitude (Fig. 14c). In the western part of the leg, the dusty layer was forecast by Meso-NH (Fig. 14e,f), with an AOD reaching almost 1, though less than the observed values of 1.2–1.5. The dusty layer forecast by Meso-NH was indeed too thin. In the eastern part of the leg, all the models failed to represent the near-surface dust layer. AROME did a better job by forecasting a dust extinction above 1 km, the closest to the observations. AROME was initialized at 18:00 UTC the previous day and was able to forecast the development of the thunderstorms over the Atlas Mountains and the associated cold pools (not shown). ALADIN did not forecast any density current, which explains why the magnitude of dust extinction was too low. The MNH05 simulation on 20 June, started at 00:00 UTC forecast thunderstorms over the Atlas





Mountains (Fig. 5). However, the meteorological imprint of the associated density currents was
removed when initializing the Meso-NH forecasts with ECMWF 00:00 UTC analysis on 21 June.

### 5.4 Afternoon planetary boundary layer in the heat low region

A last example is taken from the afternoon survey of the SABL across the heat low on 22 June.
Engelstaedter et al. (2015) analysed the meteorological situation for that day in some detail. They
showed that the monsoon flow encircled the heat low from its eastern flank, at both low- and mid-
level altitudes.

The leg of the flight F25 track between 16:01 and 17:00 UTC overlaid on the AOD field from the
satellite retrievals and the forecasts is shown in Fig. 15. According to the mean sea level pressure
(MSLP) forecasted by Meso-NH, the heat low exhibited an elongated pattern with a NE-SW orien-
tation. Its centre, located around $18°$ N/$6°$ W was sampled by the aircraft in the southern part of the
leg. In the Meso-NH forecasts, the location of the MSLP minimum was much better defined than
in the other two forecasts. The forecasts agreed on the broad structure of the wind flow at 925 hPa
characterized by a Mediterranean surge from the northeast and the monsoon flow from the south-
west. It resulted in an AOD larger than 1 over Central Mauritania (around $20°$ N/$12°$ W) for all the
forecasts. These values contrasted with the AOD of less than 0.6 from MODIS. The forecasts also
matched the AERONET value of 0.9 at Zouerate.

The vertical distribution of extinction observed by LNG showed a deep, dusty SABL (Fig. 16),
extending up to an altitude of 6 km with extinction larger than $0.4\,\mathrm{km}^{-1}$ in the first kilometre. In
the northernmost 450 km, the upper boundary layer was cloudy with little dust extinction, less than
$0.1\,\mathrm{km}^{-1}$. In the southernmost 300 km, which crossed the centre of the heat low, there was no cloud
and the dust extinction was larger than $0.1\,\mathrm{km}^{-1}$ up to an altitude of 6 km. AROME overestimated
the dust extinction in most parts of the leg. This drawback was linked to the mislocation of the heat
low centre. ALADIN, MNH20 and MNH05 forecast the dust increase towards the heat low centre
well. However, ALADIN forecast too low an extinction while Meso-NH overestimated the northern
limit of the deep, dusty layer at a distance of about 350 km (against the observed 450 km).

### 6 Conclusions

During the 2011 Fennec field campaign, the Saharan atmosphere was probed using ground-based and
airborne observations. For the purpose of aircraft guidance, dust forecasts were produced specifically
for Fennec using three limited-area models: ALADIN, AROME and Meso-NH. Among the four sets
of forecasts, two were made with a horizontal grid spacing of 5 km permitting the deep convection to
be represented explicitly. The unique data set allowed the first ever intercomparison of dust forecasts
over the western Sahara.



At monthly time scale, less AOD was forecast by the low-resolution models. By construction, these models generated lower near-surface wind speeds, which resulted in a weaker dust emission. This contrasted with the high-resolution models that forecast a much stronger variability and intensity of AOD. This was partly due to the generation of density currents that mobilized dust. This effect was well illustrated by the change in AOD and emission between the two sets of the Meso-NH forecasts. Over the western Sahara, the high-resolution one emitted more dust in the evening than its low-resolution counterpart while using a reduced sandblasting efficiency. Another striking difference was the AOD forecast over the Grand Erg Occidental. The harmattan was more prevalent there in the Meso-NH forecasts starting with ECMWF analysis than in ALADIN and AROME, which used the ARPEGE forecasts.

The agreement found with the MISR retrieval in the strip of large AOD around $18°$ N suggests that the high-resolution models performed better than the low-resolution ones. However, the MISR mean AOD differed strongly from the MODIS one, the latter showing a strip of large AOD around $14°$ N. The larger number of observations obtained by MODIS (about 28 against 5 for MISR) should give more confidence on the MODIS product. Although the MODIS AOD retrievals and the AERONET observations were in agreement over Tamanrasset, they differed significantly over Zouerate. Too large an AOD was retrieved there, while the Atlantic inflow characterized by a low aerosol load led to near-zero AOD values. The fact that this synoptic-scale feature was forecast well suggests that the MODIS retrieval was not reliable at Zouerate and certainly over some other areas. It is also worth noting that the models presented correlation coefficient values against AERONET observations that were higher than for the MODIS retrieval at Zouerate and Bordj Mokhtar.

At daily time scales, the vertical structure of temperature and humidity was forecast well. Both the mixed layer air in the SABL and the dryness of the Saharan air near the surface were captured successfully. All the models forecast dust extinction within the SABL correctly. Because the dust emission was too low, the magnitude of dust extinction forecast by ALADIN was also too low. AROME missed the larger amount of dust extinction in the first kilometre. In contrast, the Meso-NH model provided a much more realistic vertical distribution of dust extinction. The high resolution also showed fine vertical gradients of dust that appeared very realistic when compared with lidar observations.

This intercomparison underlined the importance of density currents for dust emission, which was observed in a few cases with LNG. While cold pools contributed to $6\%$ of the total dust emission in MNH05 over the western Sahara only, they dominated the dust emission over the western fringes of the Adrar des Iforas and Aïr Mountains. On the one hand, the study emphasizes the need for the convection-parameterizing models to represent this process. On the other hand, the convection-permitting models can overestimate the wind speed strongly, hence the dust emission. This issue raises some questions on the representation of subgrid eddies and associated drawbacks in the tur-



bulence scheme. Tests of sensitivity to model resolution for some dedicated case studies should be carried out in order to further investigate this issue.

*Acknowledgements.*   The Fennec-France project was funded the Agence Nationale de la Recherche (ANR 2010 BLAN 606 01), the Institut National des Sciences de l'Univers (INSU/CNRS) through the LEFE program, the Centre National d'Etudes Spatiales (CNES) through the TOSCA program, and Météo-France. Airborne data were obtained using the Falcon 20 Environment Research Aircraft operated and managed by Service des Avions Français Instrumentés pour la Recherche en Environnement (SAFIRE, www.safire.fr), which is a joint

entity of CNRS, Météo-France, and CNES. We thank Emilio Cuevas-Agullo and Martin Todd, and their staff for establishing and maintaining the AERONET sites used in this study. MODIS and MISR data were obtained from the Giovanni online data system (NASA GES DISC).



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



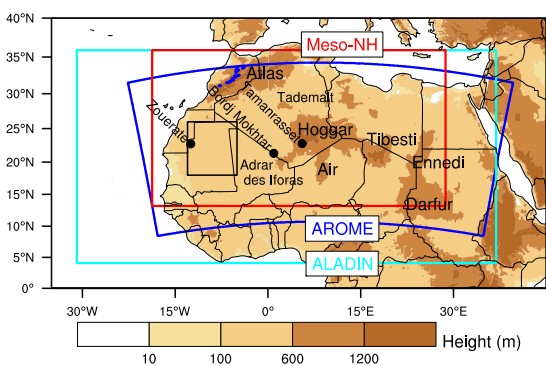

**Figure 1.** Topography of North Africa. The locations of the domains of ALADIN, AROME and Meso-NH are indicated with cyan, blue and red lines, respectively. The black rectangle shows the area of interest where most of the SAFIRE Falcon 20 data were acquired. The location of the three AERONET stations used in this study are shown with black dots.

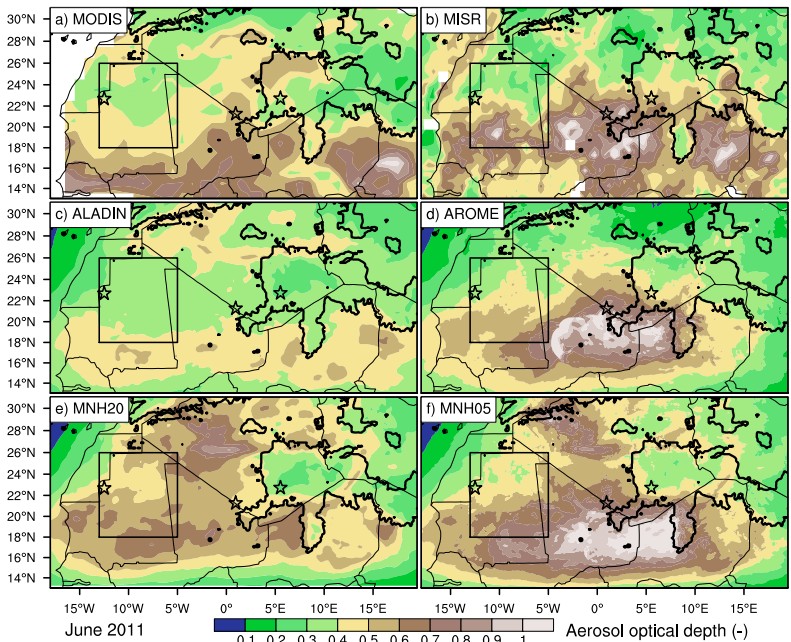

**Figure 2.** Aerosol optical depth around 12:00 UTC in June 2011 from a) MODIS, b) MISR, c) ALADIN, d) AROME, e) MNH20 and f) MNH05. The solid line shows the 600-m altitude. The black rectangle shows the domain of analysis. The locations of the three AERONET stations used in this study are shown with stars.





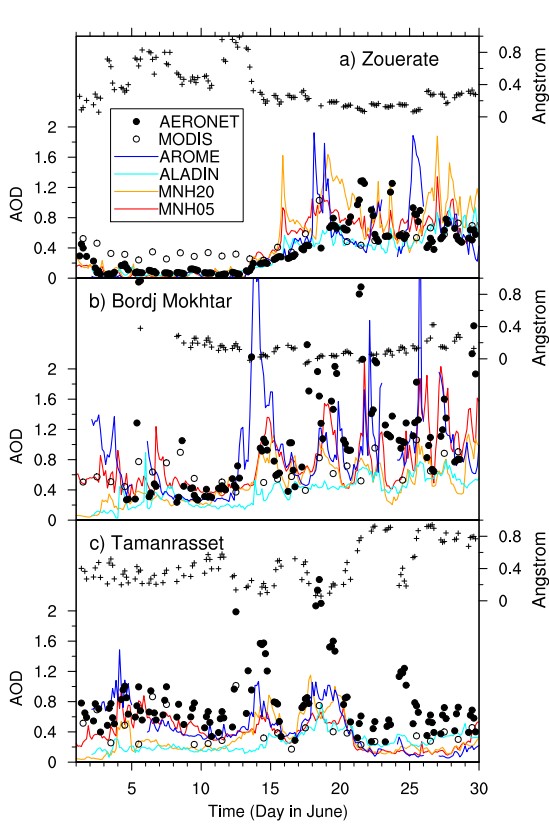

**Figure 3.** Time evolution of AOD (left vertical axis) from AERONET, MODIS, ALADIN, AROME, MNH20 and MNH05 (see legend box) and of Ångstrom exponent (right vertical axis) from AERONET (crosses) at a) Zouerate, b) Bordj Mokhtar and c) Tamanrasset.



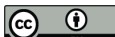

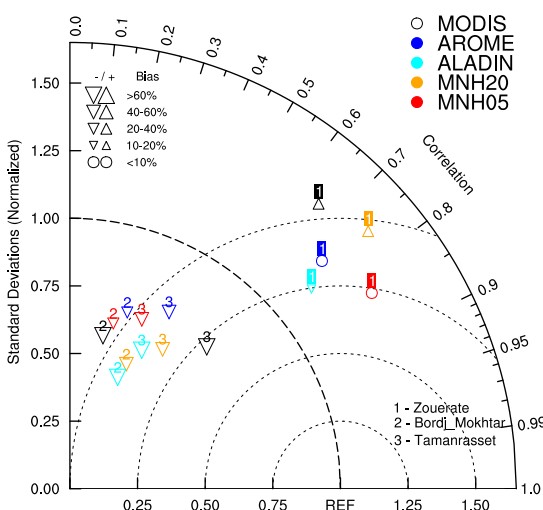

**Figure 4.** Taylor diagram showing normalized standard deviations (radius) and correlation (cosine of angle) of AOD with respect to that of the AERONET observations. The size of symbols varies with the bias relative to AERONET AOD values averaged at each station. For each model, the station number in which the bias is minimum is set in a box.

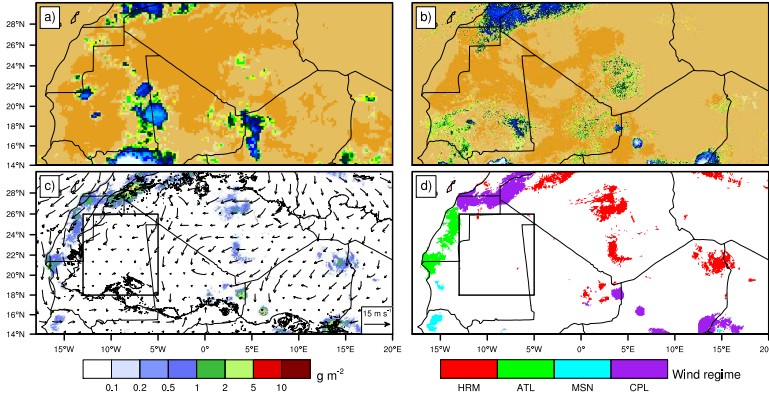

**Figure 5.** Fields at 21:00 UTC 20 June 2011. Top row: brightness temperature a) from SEVIRI observation and b) from MNH05. Bottom row: c) 3-hourly dust emission and d) attributed wind regime from MNH. HRM, ATL, MSN and CPL stand for harmattan, Atlantic flow, monsoon flow and cold pool, respectively. In c), the line represents the mixing ratio of water vapour at 2 m equal to $10\,\mathrm{g\,kg^{-1}}$ and the arrows the wind field at 2 m. In c) and d), the black rectangle shows the domain of analysis.



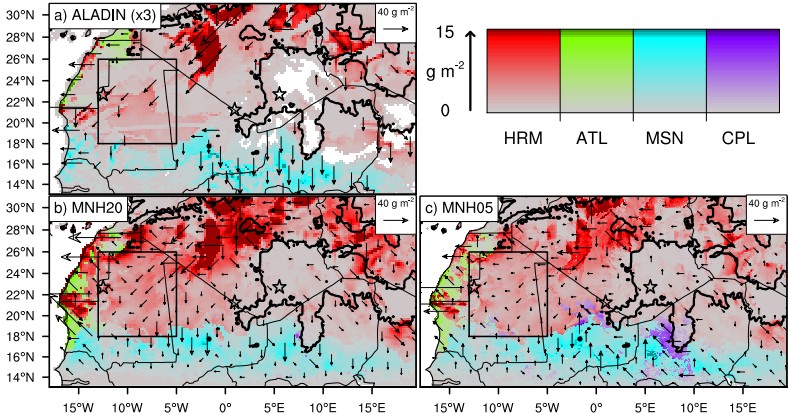

**Figure 6.** Three-hourly dust emission in June 2011 from a) ALADIN, b) MNH20 and c) MNH05. The dust emission in ALADIN was multiplied by a factor 3. The colour represents the dominant wind regime in emission and the colour intensity the magnitude of the dust emission. HRM, ATL, MSN and CPL stand for harmattan, Atlantic flow, monsoon flow, and cold pool, respectively. Arrows indicate the amplitude and phase of the diurnal variation of dust emission. The length of an arrow represents the magnitude of the maximum 3-hourly dust emission and its direction the time of the day when this maximum occurred with respect to a 24-hour clock. Arrows pointing upward indicate a peak at 00 UTC, toward the right indicate a peak at 06 UTC, etc. Arrows representing less than $2\,\mathrm{g\,m^{-2}}$ have been omitted. The solid line shows the 600-m altitude. The location of the three AERONET stations used in this study are shown with stars.



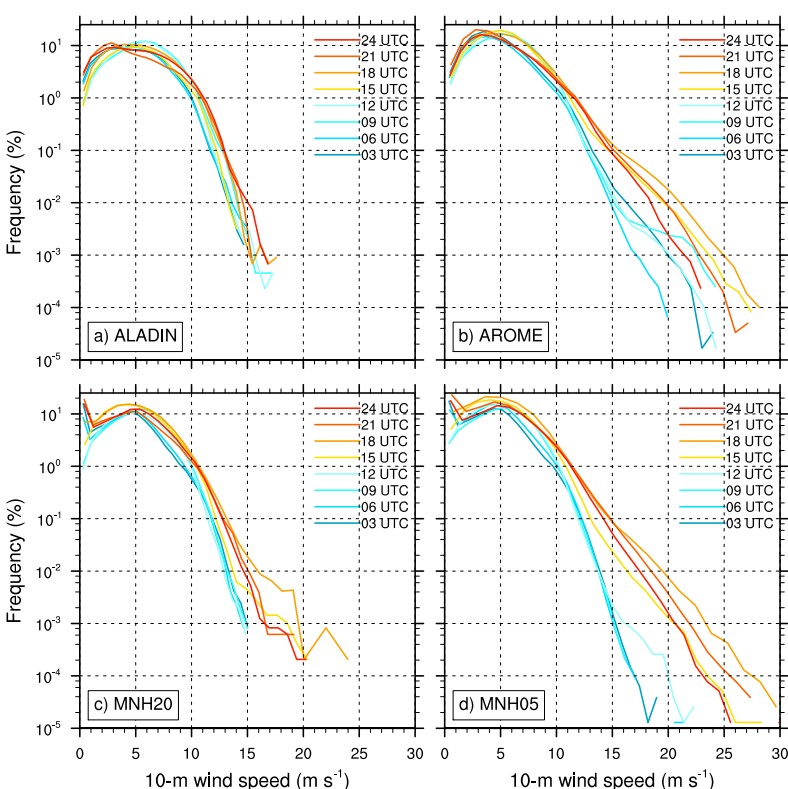

**Figure 7.** Probability density function of the 10 m wind speed in June 2011 from a) ALADIN, b) AROME, c) MNH20 and d) MNH05.

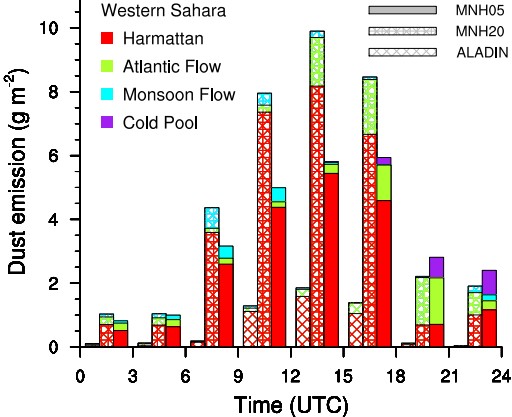

**Figure 8.** Diurnal cycle of dust emission in June 2011 from ALADIN, MNH20 and MNH05 over the western Sahara, the area of interest shown in Fig. 6.





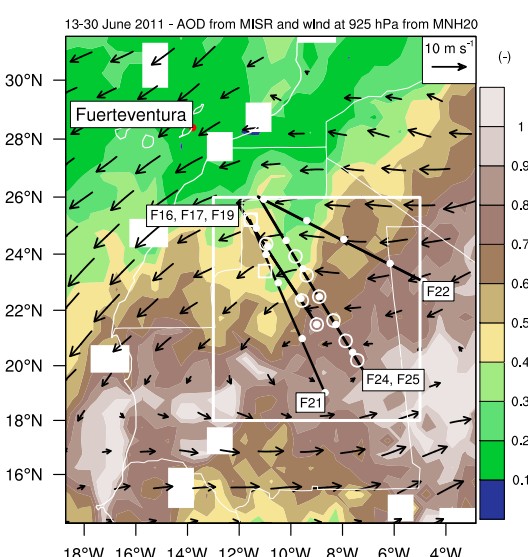

**Figure 9.** Aerosol optical depth from MISR and wind at 925 hPa from MNH20 during the heat low phase (13–30 June 2011). The white rectangle shows the area of interest and the black lines the location of the selected legs of the Falcon flight tracks shown in Fig. 10. The positions of the dropsondes are shown with dots for F16, F21, F22, and F24, open circles for F17 and F25, and squares for F19.



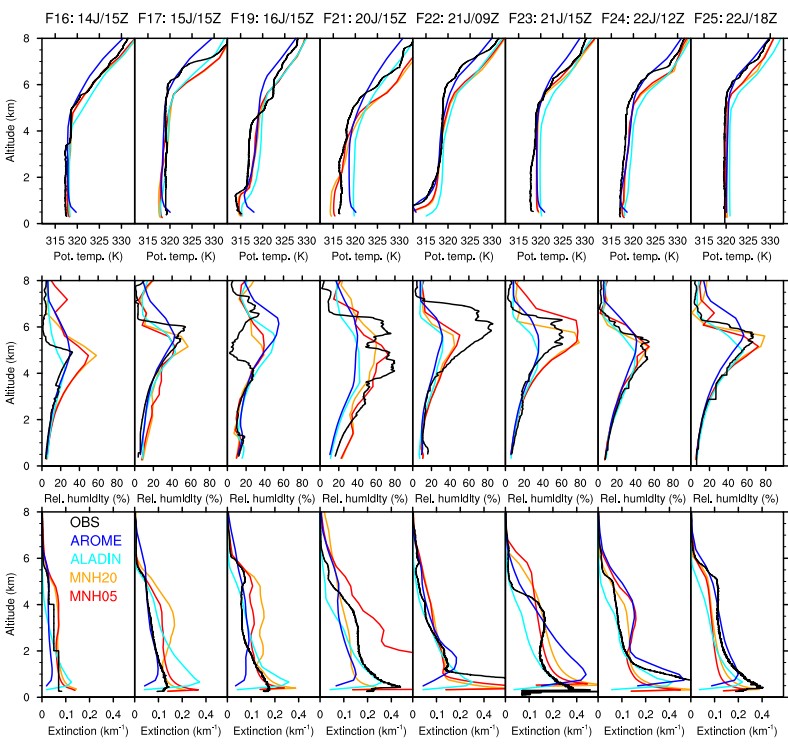

**Figure 10.** Profiles of (top) potential temperature, (middle) relative humidity and (bottom) dust extinction for the selected legs of the Falcon flights F16, F17, F19, F21, F22, F23, F24 and F25. The locations of the legs are shown in Fig. 9.





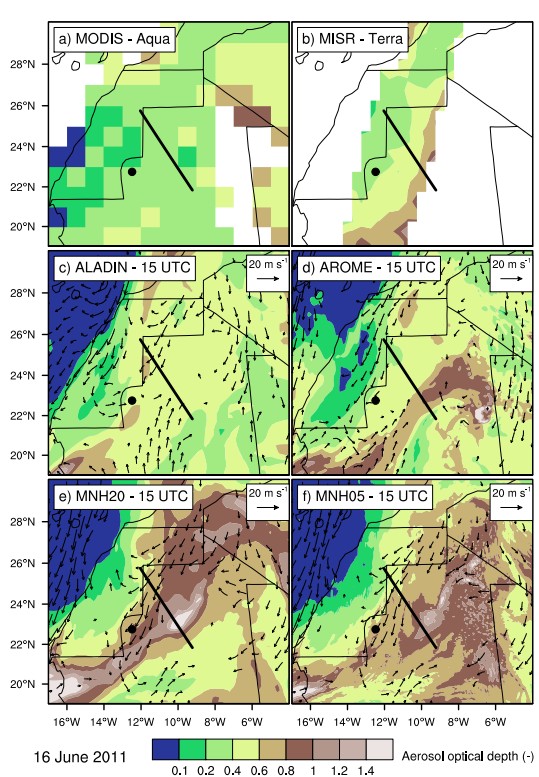

**Figure 11.** Aerosol optical depth on 16 June 2011 from a) MODIS, b) MISR, c) ALADIN, d) AROME, e) MNH20 and f) MNH05. The black line shows the location of the Falcon flight F19 track (leg between 15:18 and 16:00 UTC). The vectors show the wind at 925 hPa for speeds higher than $10\,\mathrm{m\,s^{-1}}$ forecast by the models.





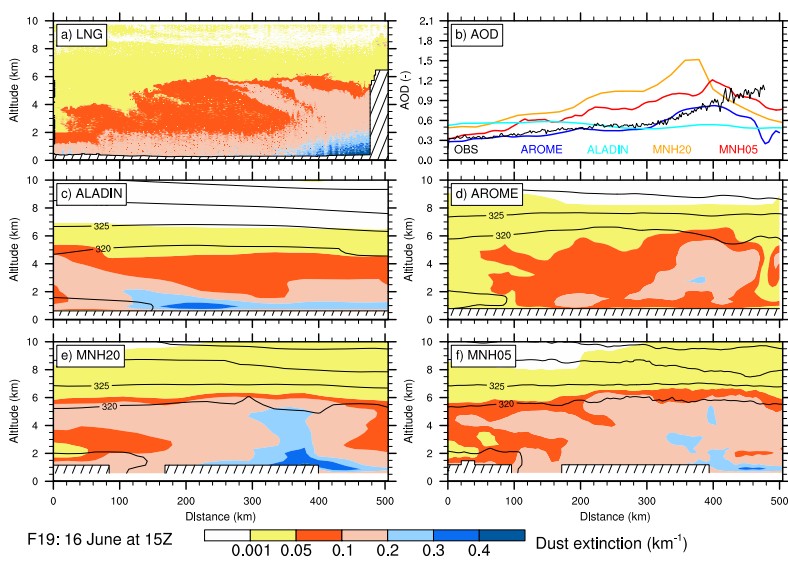

**Figure 12.** Vertical cross-section of the extinction on 16 June 2011 along the F19 track shown in Fig. 11 from a) LNG, c) ALADIN, d) AROME, e) MNH20 and f) MNH05. LNG observations were taken between 15:18 and 16:00 UTC and forecasts are at 15:00 UTC. The black lines show the potential temperature every 5 K in (c–e). Vertical hatched areas in the LNG observations are missing data due to the presence of clouds at the top of the SABL. b) Evolution of the AOD along the F19 track from LNG observations (thin black solid line) and the four forecasts (see legend).





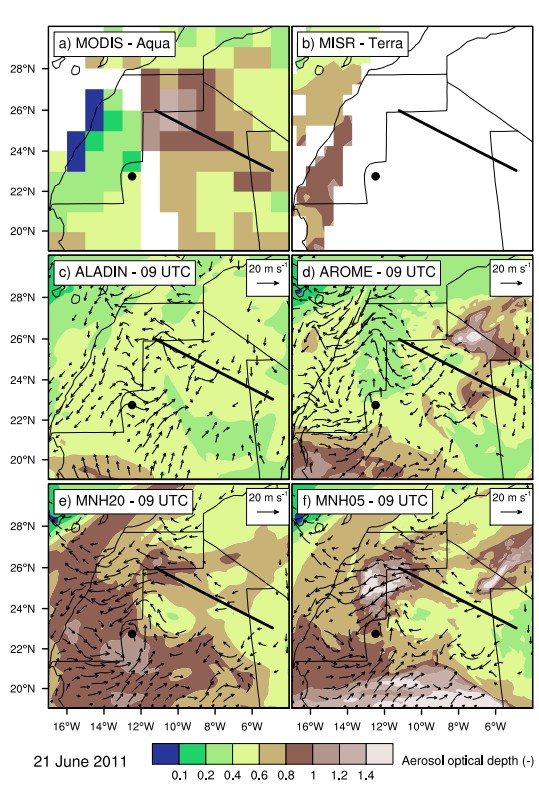

**Figure 13.** Same as in Fig. 11 but for the F22 flight on 21 June 2011 (leg between 07:58 and 08:50 UTC).





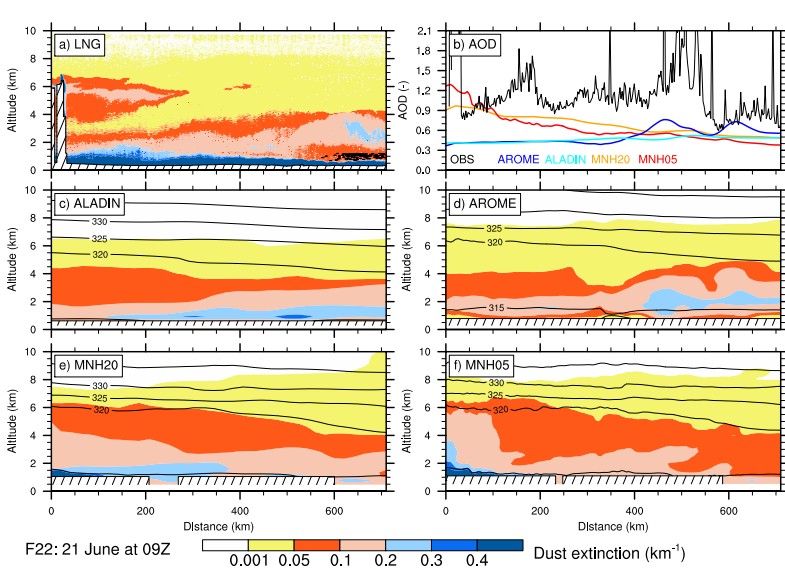

**Figure 14.** Same as in Fig. 12 but for the F22 flight on 21 June 2011. LNG observations were taken between 07:58 and 08:50 UTC and forecasts are at 09:00 UTC.





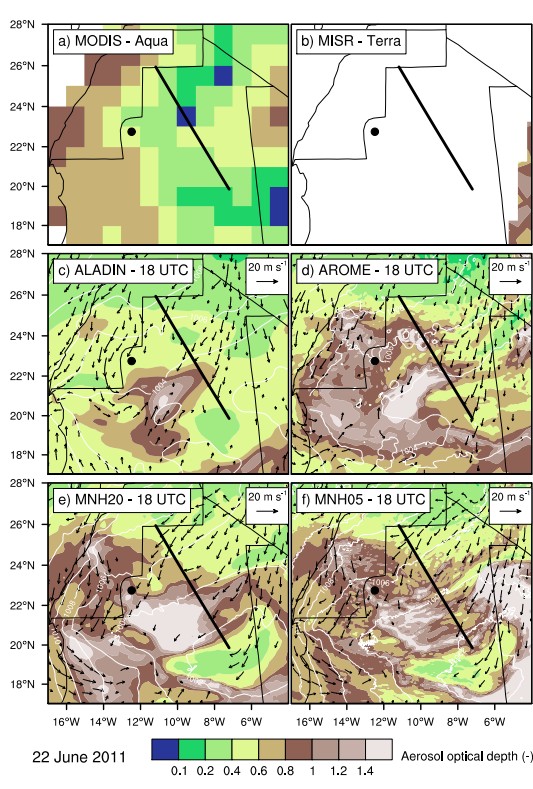

**Figure 15.** Same as in Fig. 11 but for the F25 flight on 22 June 2011 (leg between 16:01 and 17:00 UTC). White lines show MSLP in (c-f).





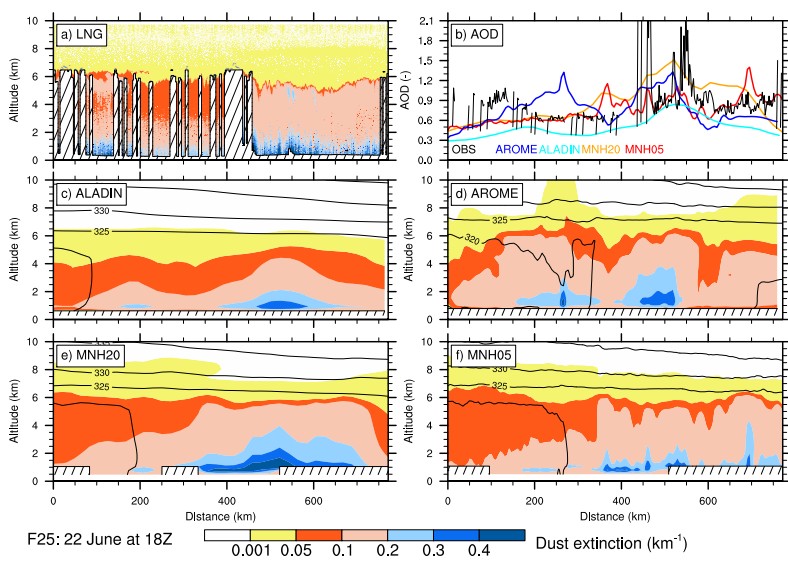

**Figure 16.** Same as in Fig. 12 but for the F25 flight on 22 June 2011. LNG observations were taken between 16:01 and 17:00 UTC and forecasts are at 18:00 UTC.