# Peer review of "Fennec dust forecast intercomparison over the Sahara in June 2011"

_Atmospheric Chemistry and Physics, 2016_

## Referee Comment (RC1) · Anonymous Referee #1 · 8 Feb 2016

This manuscript tries to look for any potential systematic error in the forecasts through evaluating the ability of three models to reproduce the key processes for mobilization and transport. The dust evaluation was made using satellite and ground-based observations.

**General comments:**

This study describes the first ever intercomparison of dust forecasts over the western Sahara by using the airborne and ground-based data sets. The result presents that, at monthly scale, large AODs were forecast over the Sahara, a feature observed by some satellite retrievals but mislocated by others over the Sahel. The AOD was correctly predicted by the high-resolution models while underestimated by the low-resolution models. The results are reasonable. However, there are ambiguous descriptions in the paper. I recommend publication after the revisions.

**Specific comments:**

1. P4, L109, 'variables at the end of a given 24 h forecast are were passed on as initial conditions at the start of', 'were' or 'are'?

2. P4, L101-109, The initial and boundary conditions of ALADIN and AROME were taken from operational large-scale ARPEGE forecasts at 18:00 UTC, while two models with different resolution of Meso-NH were initialized by the ECMWF analysis at 00:00UTC. Why don't use the same initial and boundary conditions?

3. To discrepancy of different models' result, what is the main reason? How about the contributions of initial and boundary condition and the DEAD version?

4. P18, Lin595: What is the reason of the AOD difference between MODIS and MISR? Just because of the number of observations? How about the `contribution of` retrieval algorithm?

5. The aerosol emission field is not needed to input for models? It is calculated by model itself?

6. What is the height of wind filed used to calculate the dust emission? Same or not in three models?

7. As shown in the paper, the simulation abilities of three models are different even over same area. How about the land surface situation in each model? Same or different?

8. As shown in L68 on P3, 'The objectives of this intercomparison were to look for any potential systematic error in the forecasts...', How to ensure the error is systematic and how to eliminate `such error in forecasts reasonably`?

9. On the research of dust, there are many studies on the dust property and transport basing on the observation and simulation over the world, such as:

- Kaufman, Y. J., Tanré, D., Dobocik, O., Karnieli, A., and Remer, L. A.: Absorption of sunlight by dust as inferred from satellite and groundbased remote sensing, Geophys. Res. Lett., 28, 1479–1482, 2001.
- Takemura, T., Uno, I., Nakajima, T., Higurashi, A., and Sano, I.: Modeling study of

long-range transport of Asian dust and anthropogenic aerosols from East Asia, Geophys. Res. Lett., 29, 2158, doi:10.1029/2002GL016251, 2002.

- Chen, S., J. Huang, C. Zhao, Y. Qian, R. Leung, and B. Yang, 2013: Modeling the transport and radiative forcing of Taklimakan dust over the Tibetan Plateau: A case study in the summer of 2006, Journal of Geophysical Research: Atmospheres, 118, doi:10.1002/jgrd.50122.
- Bi, J., J. Huang, Q. Fu, X. Wang, J. Shi, W. Zhang, Z. Huang, and B. Zhang, Toward characterization of the aerosol optical properties over Loess Plateau of Northwestern China, Journal of Quantitative Spectroscopy & Radiative Transfer, 112 (2) (2011), 346-360.
- Liu, Y., J. Huang, G. Shi, T. Takamura, P. Khatri, J. Bi, J. Shi, T. Wang, X. Wang, and B. Zhang, Aerosol optical properties and radiative effect determined from sky-radiometer over Loess Plateau of Northwest China, Atmospheric Chemistry and Physics, 11 (22) (2011), 11455-11463, doi:10.5194/acp-11-11455-2011.
- Huang, J., W. Zhang, J. Zuo, J. Bi, J. Shi, X. Wang, Z. Chang, Z. Huang, S. Yang, B. Zhang, G. Wang, G. Feng, J. Yuan, L. Zhang, H. Zuo, S. Wang, C. Fu and J. Chou, An overview of the semi-arid climate and environment research observatory over the Loess Plateau, Advances in Atmospheric Sciences, 25 (6) (2008), 906-921, doi:10.1007/s00376-008-0906-7.
- Huang, J., J. Ge, and F. Weng, 2007: Detection of Asia dust storms using multisensor satellite measurements, Remote Sensing of Environment, 110, 186-191.
- Huang, J., P. Minnis, Y.Yi, Q.Tang, X. Wang, Y. Hu, Z. Liu, K. Ayers, C. Trepte, and D. Winker, 2007: Summer dust aerosols detected from CALIPSO over the Tibetan Plateau, Geophys. Res. Lett., 34, L18805, doi:10.1029/2007GL029938.

Please cite above researches and inter-compare with your study.

---

## Referee Comment (RC2) · Anonymous Referee #2 · 14 Apr 2016

General Comments:

The paper presents a dust modeling intercomparison against measurements performed during Fennec campaign in June 2011. The authors focus on the ability of four limited area models (ALADIN, AROME, Meso-NH20km, Meso-NH5km) to forecast atmospheric and dust properties during the campaign, performed over the western Sahara. They claim that this is to be "the first ever intercomparison of dust forecasts over the western Sahara". Obviously this is not true since a big number of dust model evaluation studies as well as operational products for this area are already available based on comparisons with satellite and AERONET data (e.g. SDS-WAS). The possible added value in this work would come from a potential use of the field aircraft measurements for model evaluation. This is partially done in this paper but a larger part of the work is devoted on comparisons with satellites and AERONET.

Furthermore, I am concerned about the low modeling skill that is found in this study which may be due to the inability of the specific models to reproduce the atmospheric dust processes or due to improper configuration of the simulations.

Therefore I would recommend the publication of this paper, after the authors proceed with some major revisions that include:

1. Rerunning the models to homogenize initial conditions, domain setups and increase the forecasting horizon for the high resolution runs.

2. Updating satellite products used for the evaluation (e.g. MODIS collection 6, including DB for Terra)

3. Moreover I would suggest that the authors will focus also on the in-situ data provided for specific case studies of particular interest by Ryder et al. (2015). Such analysis could provide more insight on the capabilities and restrictions of each model in reproducing the dust related processes.

Details on the aforementioned suggestions are given below along with other minor comments.

Specific Comments:

Line 16: Please replace extinction with concentrations

Line 53: Two aircrafts

Line 91: centred – centered

Line 92: What is a "piecewise parabolic method advection scheme" ?

Line 103: Please give more information on ARPEGE model

Line 103: There is no use in comparing models with different initial conditions. Even the same model will produce different results when initialized at different times.

Line 106: Don't you assume any model spin up time for Meso-NH? By initializing the

model every 24 hours you are most probably just interpolating ECMWF data on your domain rather than allowing the higher resolution model to develop its own atmospheric fields.

Lines 100-115: Please give also information for the vertical model grid structure.

Lines 137-140: What is the physical meaning of this tuning factor?

Line 162: "In general, dust is characterized by low values of Ångstrom exponent, less than 0.4". Did you filter out AOD measurements with Angstrom > 0.4?

Lines 169-172 : Do you consider using updated MODIS DB product from Collection 6 for your comparisons throughout the paper? And please check if Terra DB product is available already.

Line 175: "63% of the MISR AODs fell within 0.05 or 20% of AERONET AODs". This phrase is not clear - please explain further.

Lines 233 – 307: Comparison between models and AERONET in section 3.2 indicate a very low modeling skill which makes the analysis presented in the following sections questionable.

Line 297: "There was no strong contrast in scores between forecasts initialized with ARPEGE and ECMWF nor in forecasts at low and high resolutions". How do you explain this taking also into consideration the model differences mentioned in lines 309-315?

Lines 314-315. I would strongly recommend to run again AROME for Fennec period extending also the modeling domain so that it is comparable to ALADIN.

Lines 334-335: Why not attributed to LLJ?

Line 340: "Here, the ITD was defined as the southern limit where the mixing ratio of water vapour at 2m equals 10 g kg$-1$". At which model?

[Figure]

Line 355: Please examine if these are indeed cold pools and explain how they are defined in the models.

Line 372-373 : Not clear - Please rephrase.

Lines 410-420: What is the correlation between the wind regimes and dust episode severity? For example if the 6% cold pool corresponds to one or two episodes during the 1 month period this could be a significant contribution.

Lines 457-458: How do you define "rather well" and "remarkably well"? Please use statistical metrics.

Lines 545-546: "AROME was initialized at 18:00UTC the previous day and was able to forecast the development of the thunderstorms over the Atlas Mountains and the associated cold pools (not shown)". This is an important finding and it is definitely worth showing.

Lines 549-550: "However, the meteorological imprint of the associated density currents was removed when initializing the Meso-NH forecasts with ECMWF 00:00UTC analysis on 21 June." Following my previous comment in Line 106 I would strongly recommend that you run again at least Meso-NH 5km for the Fennec period but this time including a sufficient spin up time.

Line 593: " the high-resolution models performed better than the low-resolution ones". I guess that we know that already.

Line 603: "At daily time scales, the vertical structure of temperature and humidity was forecast well". How do you define well? Please use statistical metrics.

Line 605: "All the models forecast dust extinction within the SABL correctly". So why do you multiply ALADIN emission by a factor of 3?

---

## Author Comment (AC1) · 23 May 2016

We thank the Referee for his/her time and his/her constructive comments. We have complied with most of the proposed changes. In the revised version of the manuscript, we now thank the referees explicitly in the Acknowledgment section. In the following, our point by point replies to the Referee's comments are in blue.

This manuscript tries to look for any potential systematic error in the forecasts through evaluating the ability of three models to reproduce the key processes for mobilization and transport. The dust evaluation was made using satellite and ground-based observations.

General comments:

This study describes the first ever intercomparison of dust forecasts over the western Sahara by using the airborne and ground-based data sets. The result presents that, at monthly scale, large AODs were forecast over the Sahara, a feature observed by some satellite retrievals but mislocated by others over the Sahel. The AOD was correctly predicted by the high-resolution models while underestimated by the low-resolution models. The results are reasonable. However, there are ambiguous descriptions in the paper. I recommend publication after the revisions.

Specific comments:

1. P4, L109, 'variables at the end of a given 24 h forecast are were passed on as initial conditions at the start of', 'were' or 'are'? Were. Thanks

2. P4, L101-109, The initial and boundary conditions of ALADIN and AROME were taken from operational large-scale ARPEGE forecasts at 18:00 UTC, while two models with different resolution of Meso-NH were initialized by the ECMWF analysis at 00:00UTC. Why don't use the same initial and boundary conditions? The forecasts were done on two different computer centers, one at Météo-France, the other at Laboratoire d'Aérologie. The centers had different avalaibility to computing ressources and to access to either ARPEGE forecasts or ECMWF analyses.

3. To discrepancy of different models' result, what is the main reason? How about the contributions of initial and boundary condition and the DEAD version? Indeed, the initial and boundary conditions and the different DEAD version are a source of discrepancy between the forecasts. This is discussed in the section dedicated to the AOD comparison.

4. P18, Lin595: What is the reason of the AOD difference between MODIS and MISR? Just because of the number of observations? How about the contribution of retrieval algorithm? In addition to the sampling issue, the way the AOD was retrieved is another reason of the AOD difference. MODIS Deep blue algorithm is based on observations in the blue wavelengths of the visible spectrum (412 and 470 nm) while MISR uses

four narrow spectral bands centered at 446, 558, 672, and 866 nm and nine distinct zenith angles). AOD products are then referenced to 550 and 558 nm for MODIS and MISR, respectively. Banks et al. (RSE 2013) have also shown that, over North Africa and during the Fennec campaign, AODs retrievals from these sensors were sensitive to meteorological conditions as well as to the emissivity of underlying surfaces. This information is now included in the revised version of the manuscript.

5. The aerosol emission field is not needed to input for models? It is calculated by model itself? The emission field is a diagnostic computed from the wind field and the surface characteristics at every time step and for every model mesh over land. This field was saved in the course of the ARPEGE and Meso-NH runs. Unfortunately, it was computed but not saved as an output field in the AROME simulations.

6. What is the height of wind filed used to calculate the dust emission? Same or not in three models? The wind field used for calculating the dust emission is the one at the lowest level. The lowest level differs between each model. Note that ARPEGE and AROME used a terrain-following vertical grid in pressure coordinates while it is in altitude coordinates for Meso-NH.

7. As shown in the paper, the simulation abilities of three models are different even over same area. How about the land surface situation in each model? Same or different? For all three models, the topography is derived from the GTOPO30 database, the soil characteristics are obtained from the FAO database and the vegetation is from the ECOCLIMAP database. So the land surface characteristics of each model are derived from the same databases. However they differ between the models as a result of the interpolation of these gridded databases at the horizontal resolution of the models.

8. As shown in L68 on P3, 'The objectives of this intercomparison were to look for any potential systematic error in the forecasts...', How to ensure the error is systematic and how to eliminate such error in forecasts reasonably? We did not find any systematic errors, except the low bias in the model-derived AOD over Tamanrasset. In that case,

we hypothesize that a process is not included/reproduced in the DEAD scheme, i.e. the remobilization of dust transported from remote sources, which is frequently observed in this region. We believe that improving this aspect of the DEAD scheme could improve dust forecasts, not only in northern Africa.

9. On the research of dust, there are many studies on the dust property and transport basing on the observation and simulation over the world, such as:

Kaufman, Y. J., Tanré, D., Dobocik, O., Karnieli, A., and Remer, L. A.: Absorption of sunlight by dust as inferred from satellite and groundbased remote sensing, Geophys. Res. Lett., 28, 1479–1482, 2001.

Takemura, T., Uno, I., Nakajima, T., Higurashi, A., and Sano, I.: Modeling study of long-range transport of Asian dust and anthropogenic aerosols from East Asia, Geophys. Res. Lett., 29, 2158, doi:10.1029/2002GL016251, 2002.

Chen, S., J. Huang, C. Zhao, Y. Qian, R. Leung, and B. Yang, 2013: Modeling the transport and radiative forcing of Taklimakan dust over the Tibetan Plateau: A case study in the summer of 2006, Journal of Geophysical Research: Atmospheres, 118, doi:10.1002/jgrd.50122.

Bi, J., J. Huang, Q. Fu, X. Wang, J. Shi, W. Zhang, Z. Huang, and B. Zhang, Toward characterization of the aerosol optical properties over Loess Plateau of Northwestern China, Journal of Quantitative Spectroscopy & Radiative Transfer, 112 (2) (2011), 346-360.

Liu, Y., J. Huang, G. Shi, T. Takamura, P. Khatri, J. Bi, J. Shi, T. Wang, X. Wang, and B. Zhang, Aerosol optical properties and radiative effect determined from sky-radiometer over Loess Plateau of Northwest China, Atmospheric Chemistry and Physics, 11 (22) (2011), 11455-11463, doi:10.5194/acp-11-11455-2011.

Huang, J., W. Zhang, J. Zuo, J. Bi, J. Shi, X. Wang, Z. Chang, Z. Huang, S. Yang, B. Zhang, G. Wang, G. Feng, J. Yuan, L. Zhang, H. Zuo, S. Wang, C. Fu and

J. Chou, An overviewthe semi-arid climate and environment research observatory over the Loess Plateau, Advances in Atmospheric Sciences, 25 (6) (2008), 906-921, doi:10.1007/s00376-008-0906-7.

Huang, J., J. Ge, and F. Weng, 2007: Detection of Asia dust storms using multisensor satellite measurements, Remote Sensing of Environment, 110, 186-191.

Huang, J., P. Minnis, Y.Yi, Q.Tang, X. Wang, Y. Hu, Z. Liu, K. Ayers, C. Trepte, and D. Winker, 2007: Summer dust aerosols detected from CALIPSO over the Tibetan Plateau, Geophys. Res. Lett., 34, L18805, doi:10.1029/2007GL029938.

Please cite above researches and inter-compare with your study.

We thank the reviewer for the interesting references. However, we find it difficult to include references to inter-comparison studies that took place in areas other than Northern Africa. Since the above mention studies relate to China, exception made of the Kaufman et al. paper, we have not included them as well as the Kaufman et al. paper that discussed aerosol optical properties over Dakar and Cape Verde, that is outside the area under scrutiny.

---

## Author Comment (AC2) · 23 May 2016

We thank the Referee for his/her time and his/her constructive comments. We have complied with most of the proposed changes. In the revised version of the manuscript, we now thank the referees explicitly in the Acknowledgment section. In the following, our point by point replies to the Referee's comments are in blue.

The paper presents a dust modeling intercomparison against measurements performed during Fennec campaign in June 2011. The authors focus on the ability of four limited area models (ALADIN, AROME, Meso-NH20km, Meso-NH5km) to forecast atmospheric and dust properties during the campaign, performed over the western Sahara. They claim that this is to be "the first ever intercomparison of dust forecasts over the western Sahara". Obviously this is not true since a big number of dust model
evaluation studies as well as operational products for this area are already available based on comparisons with satellite and AERONET data (e.g. SDS-WAS). The possible added value in this work would come from a potential use of the field aircraft measurements for model evaluation. This is partially done in this paper but a larger part of the work is devoted on comparisons with satellites and AERONET.

Furthermore, I am concerned about the low modeling skill that is found in this study which may be due to the inability of the specific models to reproduce the atmospheric dust processes or due to improper configuration of the simulations.

Therefore I would recommend the publication of this paper, after the authors proceed with some major revisions that include:

1. Rerunning the models to homogenize initial conditions, domain setups and increase the forecasting horizon for the high resolution runs. Even though we agree this would be interesting to do, this proposal is definitely beyond the scope of the present paper. Our aim is really to compare the real-time aerosol-related model outputs of the models in the status of development that were in at the time of the Fennec campaign. Comparing those outputs for the current state of these models (some have evolved more than others) would be an entirely different study. Hence, we have decided not to rerun the models as suggested by the referee.

2. Updating satellite products used for the evaluation (e.g. MODIS collection 6, including DB for Terra). We thank the Referee for informing us on this new MODIS collection 6. The comparison with MODIS retrievals was updated using the AOD from the collection 6 from both Aqua and Terra. These products led to improved comparisons with the other AOD retrievals. However, they are still biased low compared to the MISR retrievals. This can be seen in Figures 2, 3, 4, 11, 13 and 15. The relevant comments and conclusions were changed accordingly.

3. Moreover I would suggest that the authors will focus also on the in-situ data provided for specific case studies of particular interest by Ryder et al. (2015). Such

analysis could provide more insight on the capabilities and restrictions of each model in reproducing the dust related processes. Thanks for suggesting this. We have not included such a comparison in the paper for the following reasons. Experience shows that such comparison is difficult and not really useful for providing insight into the capability in reproducing the dust related processes at small scales. First of all, such detailed measurements acquired over relatively small areas are often difficult to compare with model outputs at the resolution of the models. In the lower half of the Saharan atmospheric boundary layer (SABL), they are generally representative of local emissions and characterized by large natural variability at the scale of the model mesh, both of these aspects being challenging for mesoscale models (operating in real-time) to capture. In the upper part of the SABL, dust composition in the Western Sahara is dominated by long-range transport and is characterized by a mixture of dust from many remote sources. In this case, in situ observations are representative "averaged" dust characteristics and generally are a better match to the model outputs, which are integrative by nature (at the model resolutions considered here). It is worth noting that the ORILAM scheme (in which the three dust lognormal modes are generated and transported, used in all three models) was thoroughly validated using in situ aerosol measurements acquired in the long-range transport region (i.e. in the Saharan Aerosol Layer) during the AMMA campaign over a long period of time and across a large domain in West Africa (including the southern fringes of the Sahara). Hence in the case of the Fennec campaign, we expect in situ measurements collected from the BAE 146 in the dust long-range transport region (above 2 km agl) to be a good match to the model outputs, and we see no worth in adding the comparison.

Details on the aforementioned suggestions are given below along with other minor comments.

Specific Comments:

Line 16: Please replace extinction with concentrations. Done

[Figure]

Line 53: Two aircrafts. The plurial of aircraft is aircraft.

Line 91: centred – centered. The text was written in British English.

Line 92: What is a "piecewise parabolic method advection scheme" ? It is a classical advection scheme, widely used in the fluid dynamics community.

Line 103: Please give more information on ARPEGE model. ARPEGE is the Météo-France global model (this is now added in the text). In other words, it is the Météo-France equivalent of IFS at ECMWF.

Line 103: There is no use in comparing models with different initial conditions. Even the same model will produce different results when initialized at different times. We do not agree. The "poor-man" ensemble approach has been shown to be extremely useful for advancing knowledge in many fields of the atmospheric science. Our aim is really to compare the real-time aerosol-related model outputs of the models in the status of development that were at the time of the Fennec campaign. That means having to cope with different resolutions, different initial states and different parametrizations.

Line 106: Don't you assume any model spin up time for Meso-NH? By initializing the model every 24 hours you are most probably just interpolating ECMWF data on your domain rather than allowing the higher resolution model to develop its own atmospheric fields. We used three-hourly outputs starting from 3-h lead time for Meso-NH. So we implicitly assumed a spin-up time of less than 3 hours. This is a sufficient time for the model to develop its own fields.

Lines 100-115: Please give also information for the vertical model grid structure. Added.

Lines 137-140: What is the physical meaning of this tuning factor? for adjusting parameterisation ... fixing the (un)known unknown, which is a common practice in a numerical world.

Line 162: "In general, dust is characterized by low values of Ångstrom exponent, less

than 0.4". Did you filter out AOD measurements with Angstrom > 0.4? No - but both AOD and Angstrom are shown explicitly in Fig. 3, though.

Lines 169-172 : Do you consider using updated MODIS DB product from Collection 6 for your comparisons throughout the paper? And please check if Terra DB product is available already. The comparison with MODIS retrievals was updated using the AOD from the collection 6 from both Aqua and Terra.

Line 175: "63% of the MISR AODs fell within 0.05 or 20% of AERONET AODs". This phrase is not clear - please explain further. The sentence refers to Figure 2 of Kahn et al. (2005). To make the sentence more clear, we reworded it as "63% of the MISR AODs fell within the envelope of ±0.05 or ±20% × AOD of AERONET values".

Lines 233 – 307: Comparison between models and AERONET in section 3.2 indicate a very low modeling skill which makes the analysis presented in the following sections questionable. Point-by-point comparison gives low score often. This is especially true within the dust source regions.

Line 297: "There was no strong contrast in scores between forecasts initialized with ARPEGE and ECMWF nor in forecasts at low and high resolutions". How do you explain this taking also into consideration the model differences mentioned in lines 309-315? We do not expect any linear relationship between the contrast in scores and the number of differences in the model characteristics. The absence of any strong contrast in scores just indicates that the models all have good skill in forecasting dust AOD.

Lines 314-315. I would strongly recommend to run again AROME for Fennec period extending also the modeling domain so that it is comparable to ALADIN. As explained before, this is really out of the scope of the present paper. Note that the AROME domain was the largest possible given the computing capability available to us in 2011. It resulted from a compromise between computing efficiency and the necessity to cover both the dust sources and the location of the Fennec campaign.

Lines 334-335: Why not attributed to LLJ? We visually checked that a standard deviation of 10-m wind speed larger than 3 m/s corresponds well to cold pools, and not to LLJ.

Line 340: "Here, the ITD was defined as the southern limit where the mixing ratio of water vapour at 2m equals 10 g kg-1". At which model? For all the models.

Line 355: Please examine if these are indeed cold pools and explain how they are defined in the models. We checked that these structures are really cold polds, i.e. they are characterized by divergent winds and cold air at the surface.

Line 372-373 : Not clear - Please rephrase. To make the sentence more clear, we reworded it as "The time of the maximum [...] occurred at 12:00 UTC for ALADIN and MNH20 and 00:00 UTC for MNH05."

Lines 410-420: What is the correlation between the wind regimes and dust episode severity? For example if the 6% cold pool corresponds to one or two episodes during the 1 month period this could be a significant contribution. We added "Emission due to cold pools [...], but occurred 13 % of days in June (on 16, 19, 23 and 24 June)"

Lines 457-458: How do you define "rather well" and "remarkably well"? Please use statistical metrics. We added "with maximum errors of less than 4 K" after "rather well". Note that "remarkably well" was already quantified with number as the sentence is "remarkably well, with maximum differences of less than 2 K below 4 km"

Lines 545-546: "AROME was initialized at 18:00 UTC the previous day and was able to forecast the development of the thunderstorms over the Atlas Mountains and the associated cold pools (not shown)". This is an important finding and it is definitely worth showing. This will be the subject of a forthcoming study dedicated to the observations of the cold-pool properties over northern Mauritania and their representation in the models at different horizontal resolutions.

Lines 549-550: "However, the meteorological imprint of the associated density currents was removed when initializing the Meso-NH forecasts with ECMWF 00:00 UTC analysis on 21 June." Following my previous comment in Line 106 I would strongly recommend that you run again at least Meso-NH 5km for the Fennec period but this time including a sufficient spin up time. See our response above.

Line 593: " the high-resolution models performed better than the low-resolution ones". I guess that we know that already. That is not always true. A well-known example is the double-penalty effect: A high resolution forecast of the same pattern as the observations but missing the observation area scores worse than a low-resolution forecast matching partly with the observation area. In the present study, the fact that the high-resolution models performed better is an indication that these models are doing a good job at correctly pin-pointing dust emissions in both time and space, in spite of their potential for producing highly variable and more spurious fields.

Line 603: "At daily time scales, the vertical structure of temperature and humidity was forecast well". How do you define well? Please use statistical metrics. We added "with maximum errors often limited to less than 4 K and 20 %, respectively".

Line 605: "All the models forecast dust extinction within the SABL correctly". So why do you multiply ALADIN emission by a factor of 3? The wording is indeed misleading. To clarify the point, we reworded it as "All the models forecast the decrease of dust extinction with altitude within the SABL correctly."

---

## Author Response (AR2)

**1 Response to Editor**

I think you should respect to reviewer #1 for at least citing one or two he suggests reference.

Following your invitation, we added a citation to papers by Kaufman et al. (2001), Chen et al. (2013) and Liu et al. (2015).

Kaufman, Y. J., Tanré, D., Dobocik, O., Karnieli, A., and Remer, L. A.: Absorption of sunlight by dust as inferred from satellite and groundbased remote sensing, Geophys. Res. Lett., 28, 1479–1482, 2001.

Chen, S., J. Huang, C. Zhao, Y. Qian, R. Leung, and B. Yang, 2013: Modeling the transport and radiative forcing of Taklimakan dust over the Tibetan Plateau: A case study in the summer of 2006, Journal of Geophysical Research: Atmospheres, 118, doi:10.1002/jgrd.50122.

[revised manuscript text omitted]